# Integrated Omics Analyses Identify Key Pathways Involved in Petiole Rigidity Formation in Sacred Lotus

**DOI:** 10.3390/ijms21145087

**Published:** 2020-07-18

**Authors:** Ming Li, Ishfaq Hameed, Dingding Cao, Dongli He, Pingfang Yang

**Affiliations:** 1State Key Laboratory of Biocatalysis and Enzyme Engineering, School of Life Sciences, Hubei University, Wuhan 430062, China; limit@hubu.edu.cn (M.L.); hedongli@hubu.edu.cn (D.H.); 2Departments of Botany, University of Chitral, Chitral 17200, Khyber Pukhtunkhwa, Pakistan; pakoonleo@yahoo.com; 3Institue of Oceanography, Minjiang University, Fuzhou 350108, China; caodingding@wbgcas.cn

**Keywords:** sacred lotus, petiole rigidity, proteomics, cell wall, lignin biosynthesis

## Abstract

Sacred lotus (*Nelumbo nucifera* Gaertn.) is a relic aquatic plant with two types of leaves, which have distinct rigidity of petioles. Here we assess the difference from anatomic structure to the expression of genes and proteins in two petioles types, and identify key pathways involved in petiole rigidity formation in sacred lotus. Anatomically, great variation between the petioles of floating and vertical leaves were observed. The number of collenchyma cells and thickness of xylem vessel cell wall was higher in the initial vertical leaves’ petiole (IVP) compared to the initial floating leaves’ petiole (IFP). Among quantified transcripts and proteins, 1021 and 401 transcripts presented 2-fold expression increment (named DEGs, genes differentially expressed between IFP and IVP) in IFP and IVP, 421 and 483 proteins exhibited 1.5-fold expression increment (named DEPs, proteins differentially expressed between IFP and IVP) in IFP and IVP, respectively. Gene function and pathway enrichment analysis displayed that DEGs and DEPs were significantly enriched in cell wall biosynthesis and lignin biosynthesis. In consistent with genes and proteins expressions in lignin biosynthesis, the contents of lignin monomers precursors were significantly different in IFP and IVP. These results enable us to understand lotus petioles rigidity formation better and provide valuable candidate genes information on further investigation.

## 1. Introduction

Plant architecture is the embodiment of space utilization. There are numerous factors contribute to plant architecture including stem height, leaf and branching mode [1]. In crops, several factors such as stem rigidity, branching pattern, and plant height which strongly influences crop yields and efficiency of harvesting have been extensively studied [2,3]. The sacred lotus is a relic aquatic plant with two leaf types, namely the floating leaf and the vertical leaf. Floating leaf petiole exhibits flexibility and floating tendency despite it being longer than the depth of the water, while the vertical leaf petiole is rigid and erect (Figure 1). Moreover, there is no reported interconvertibility between these two kinds of leaves during the entire life span of lotus, meaning that lotus is not a heterophyllous plant. Stem or petiole rigidity is species-specific and plastic [4] controlled by environmental factors including light, temperature, water and nutrients, among others.

Cell wall makes plant cell different from animal cell. Cell wall contributes to plant architecture, organ development, transport of water and nutriment and defense. In order to accomplish variety function mentioned above, cell in plant was specialized into collenchyma, sclerenchyma and xylem [5,6,7]. Generally, cell wall can be divided into middle lamella, primary wall and secondary wall if the cell wall has secondary thickening. The main components in secondary cell wall are polysaccharide and lignin [8]. Polysaccharide mainly includes cellulose, hemicellulose and pectin. Lignin is compounded of three primary monolignols including coniferyl alcohol, sinapyl alcohol and p-coumaryl alcohol in varieties of proportions [9]. The components of cell wall are diversified depending on the plant taxa, age, tissue and cell type [8]. Studies have verified that many genes are associated with cell wall biosynthesis and secondary cell wall thickening in *Arabidopsis* [10], Populus [6] and other species [11]. Typically, cellulose synthase [12,13], glycosyltransferases and glycosyl hydrolases [8,14] play a crucial role in polysaccharide biosynthesis, and enzymes involved in phenylalanine deamination, hydroxylation, methylation and redox reactions in phenylpropanoid pathway play a key role in lignin biosynthesis [15]. Transcription factor families are engaged more in the upstream of cell wall biosynthesis and modification, including NAC domain (VND6/7:vascular-related NAC domain 6/7, SND1: secondary wall-associated NAC domain 1 and NST1: NAC secondary wall thickening promoting factor) and HD-ZIP homeobox, which were reported that they play sufficient role in influencing secondary cell wall synthesis in xylem vessel [16,17,18]. Members in MYB family are also informed as regulators involved in secondary cell wall formation [19], while members in E2F family were affirmed as a fundamental upstream transcriptional regulator of VND6/7 and other secondary cell wall biosynthesis genes [10].

Despite the increasing understanding on cell wall biosynthesis in the past two decades, a lot still remain elusive regarding related gene regulation. The findings in heterophyllous plants recommend that light and temperature are involved in transformation of heterophyllous leaves [20], and internal factors including phytohormones, such as gibberellin (GA), abscisic acid (ABA) and ethylene play a dominant role in the induction of heterophylly in plants [21]. Besides, some studies focused on *Arabidopsis* petioles indicating light quality, phytohormones and crosstalk between light and phytohormones mediate petiole elongation [22], and it is ascertained that numerous genes are involved in cell wall formation and secondary cell wall thickening in *Arabidopsis* [23], rice [24] and Populus [25]. Since there is little research done, it is unclear whether lotus has the same mechanism of gene network regulation as typical heterophyllous plants. Furthermore, lotus is the only aquatic plant which has a combination of floating and vertical leaves, and it is a good example to probe the mechanism that controls differentiation of these two leaves. These findings will facilitate us to understand better the mechanism of cell wall formation.

High-throughput profiling techniques used in transcriptome and proteome are effective in exploring complex biologic processes at the overall level [26]. The expression levels of mRNAs and proteins can be accurately measured using RNA-sequencing, iTRAQ (isobaric tags for relative and absolute quantitation) and TMT (tandem mass tag) technologies. As we known, the regulation of genes at transcriptional, translational and post-translational levels are different, integrated analysis of these data are necessary [27]. In *Arabidopsis*, transcriptome and proteome technologies were used to identified genes involved in cell wall biosynthesis. It also indicates no clear correlation was found between the omics data demonstrating complementary in different methods [28,29].

In this study, the IFP (initial floating leaves’ petiole) and IVP (initial vertical leaves’ petiole) were used to as the research material for analysis through RNA sequencing and protein labeling quantification with tandem mass tags (TMT) technology to explore the mechanism that controls differentiation of these two types of leaves in lotus. By analyzing DEGs (genes differentially expressed between IFP and IVP), DEPs (proteins differentially expressed between IFP and IVP) and the abundance of metabolite in the IFP and IVP, we found that genes highly expressed in IVP in several central biologic processes, such as cell wall biosynthesis, organization, assembly and lignin biosynthesis were associated with lotus petiole rigidity. The integrated analysis of transcriptomic and proteomic data provides a clue to understand the formation of different petioles in lotus.

## 2. Results

### 2.1. Phenotypic Evaluation of Lotus Petioles at IFP and IVP Stages

Lotus, as an herbaceous perennial plant, sprouts as temperature rises toward spring. At the beginning, several initial leaves (three to five leaves) grow and float on, but not rise above the water surface in spite of long enough petioles. Then the succeeding leaves stand out of the water (Figure 1a,b, Appendix A). Generally speaking, initial leaf floats on the water surface all the time. Vertical leaf rises above the water surface once the petiole length is larger than water depth (Figure 1c, Appendix A). Moreover, both the folded and unfolded vertical leaves can rise above the water surface (Figure 1d, Appendix A). We have not observed any interconvertibility between IFP and IVP under green house and field conditions. Numbers of lateral branches grow continuously during its growing season, and each branch develops the floating leaf first, followed by the vertical leaf.

For the purpose of quantifying the phenotype difference between the floating and vertical leaves, the breaking resistance of two type petioles was measured. The breaking resistance of IVP was significantly higher than that in IFP (Figure 1e). Furthermore, we observed two types of petioles in lotus with low water depth. The minimum length of IFP is over water depth at that condition, and IVP was greater than IFP. Even though the petiole length of floating leaf was greater than water depth as it matures, it still could not stand erect on water (Figure 1f). This result suggests the breaking resistance, but not the petiole length being the main factor affecting the leaf emergence from water surface.

### 2.2. Anatomic Structure Analysis of Petioles at IFP and IVP Stages

To explore the structure difference of petioles in floating and vertical leaf, we studied the sections of IFP and IVP. There were four primarily air cavities (the numbers of air cavities will increase as the petioles grow) in both transection of IFP and IVP. Numbers of small air cavities were scattered around the chief air cavities (Figure 2a,b), and some of them could develop into the chief air cavities along with petioles growth. The vascular bundles, which increase cell wall resistance and rigidity, were scattered around the big air cavities as well. These data showed the number of vascular bundles in IVP is more than those in IFP (Figure 2a,b,e). Furthermore, the epidermal cell had high level of cutinization in IVP than that in IFP. The outer cortex cells in IVP have specialized into collenchyma by uneven thickening of cell wall to strength the rigidity of petioles. In IFP, by contrast, the cortex cells had no cell wall thickening. (Figure 2c,d). These vascular bundles and collenchyma could contribute to cell wall resistance and rigidity thus supporting petioles emergence from water in IVP.

To further study the structure character of petiole, thickness of xylem vessels cell wall was measured (Figure 3a–d). It showed that xylem vessels cell wall in IFP was 0.4 μm compared with IVP (0.8 μm) (Figure 3g). This result suggests that xylem vessel in IVP may provide more strength to support IVP to remain upright. The main components of cell wall are polysaccharide and lignin, and the content of cellulose and lignin in cell wall were measured. This result noted that the cellulose contents were 18.7% and 24.8% in IFP and IVP, and the lignin contents were 13.3% and 21.9% in IFP and IVP, respectively (Figure 2f,g). Both cellulose and lignin contents were significantly higher in IVP than that in IFP. This result is consistent with morphologic results which displayed more vascular bundles, collenchyma and cell wall thickening in IVP compared to IFP.

### 2.3. Overview of Transcriptomic Analysis and Transcriptomic Data Validation

To explore the different genes involved in cell wall formation in IFP and IVP, comparative transcriptomic experiment was done. RNA-seq was performed for IFP and IVP groups on the Illumina platform and each group included three biologic replicates. Generally, 87–90% of the clean reads were mapped to the genome of the sacred lotus (https://bioinformatics.psb.ugent.be/plaza/versions/plaza_v4_dicots/). The expression estimates were calculated as fragments per kilobase per million reads (FPKM). Pearson’s correlation coefficient was used to evaluate repeatability of individual sample (Appendix A). A total of 20,109 genes or transcripts (about 75% predicted genes from genome) were obtained from the six samples (Figure 4a). Differentially expressed genes (DEGs) were screened based on an absolute fold change value of |log2 ratio| ≥ 1 and false discovery rate (FDR) ≤ 0.01. A total of 1422 DEGs were found, including 401 upregulated genes and 1021 downregulated genes (IVP vs. IFP) (Figure 4b, Appendix A).

The expression levels of 14 genes involved in the cell wall biosynthesis and 8 genes with different expression patterns in IFP and IVP were determined by quantitative reverse transcription polymerase chain reaction (qRT-PCR) to validate the transcriptomic data. A total of 19 genes shared the same expression pattern between transcriptomic and qRT-PCR results (Figure 4c, Appendix A). The transcriptomic data and qRT-PCR results of these genes were highly correlated (r = 0.8339; Figure 4d). These results further proved that the transcriptomic data were reliable.

### 2.4. Overview of Quantitative Proteomics Analysis and Data Validation Using PRM

To scrutinize the different proteins involved in cell wall formation of IFP and IVP, quantitative and comparative proteomics was done. A total of 25,529 peptides belonging to 5808 proteins (about 22% predicted genes from genome) were identified in the petiole samples. Among these identified proteins, 4855 (about 18% predicted genes from genome) of them were quantified (Figure 4a). Pearson’s correlation coefficient was used to evaluate repeatability of individual sample (Appendix A). After filtering with an expression fold change of >1.5 and *p*-value < 0.05, 904 out of the quantified proteins were defined as differentially expressed (DEPs) between IFP and IVP. Among these DEPs, the abundance of 421 proteins in IFP was 1.5 times higher than in IVP, and the abundance of 483 proteins in IVP was 1.5 times higher than in IFP (Appendix A).

To validate the reliability of tandem mass tag-labeling (TMT-labeling) protein quantification in this study, parallel reaction monitoring (PRM) method was used. The results showed that 11 out of 15 selected proteins were successfully quantified using PRM methods, and the changes in all 11 proteins abundances between IFP and IVP were consistent with TMT-labeling protein quantification (Appendix A). The PRM data and proteomics results of these proteins were highly correlated (r = 0.6301, Appendix A). This result confirms that our protein quantification result was reliable.

### 2.5. Integrated Analysis of Transcriptome and Proteome Data

To analyze the correlation of DEGs and DEPs identified in IFP and IVP, data from transcriptome and proteome were compared. Globally, a total of 20,109 transcripts (approximate 75.36% genome) were identified in lotus petioles, and transcripts were detected for 93.12% of the proteins (Figure 4a). To explore the relationship between protein abundance and their corresponding gene expression, we conducted a correlation analysis using Spearman’s rank correlation coefficient method between the transcriptome and quantitative proteome data. After analyzing, the expression levels of all the transcripts and their corresponding quantified proteins from IVP vs. IFP showed weak correlation (r = 0.2765, Figure 5a). The weak correlation was also observed in various multi-omics researches in *Arabidopsis*, rice and eggplant [26,30,31]. At the same time, a stronger correlation was discerned between the DEGs and their corresponding DEPs (r = 0.6148, Figure 5d). The expression ratio of proteins and their corresponding mRNAs with the same or opposite trend (both upregulated or both downregulated) were also plotted, and higher positive or negative correlation was indicated (Figure 5b,c,e,f).

In the present study, a total of 1422 DEGs and 904 DEPs were found. Both the DEGs and DEPs were classified into various functional groups (Figure 4b). Among these DEGs and DEPs, 135 of them had quantitative information both in transcript and protein level. For ease of description, the 135 genes both identified in transcriptional and translational levels were named as core genes. MapMan analysis showed the 135 core genes were classified in 22 functional groups. Except for unknown function genes, 72% of core genes were sorted into five larges groups including miscellaneous, stress, signaling, protein and cell wall (Appendix A). Among the 135 core genes, 126 of them showed same trend in transcriptional and translational level, while only 9 genes showed the opposite trend at the two levels (Appendix A). This suggests that these core genes play constant role at both RNA and protein levels in petioles development. Besides, a total of 67 transcription factors were identified in DEGs and 3 transcription factors were identified in DEPs. Only AP2/ERF and B3 domain-containing transcription factor RAV1-like both identified in DEGs and DEPs (Appendix A). Combining MapMan and GO annotation results, 11 non-redundant genes out of 135 core genes were predicted as being involved in cell wall biosynthesis, degradation and assembly (Table 1). GO and KEGG (Kyoto Encyclopedia of Genes and Genomes) pathway enrichment analysis of the 135 core genes showed that majority genes were enriched in phenylpropanoid biosynthesis and chitin catabolic process (Appendix A). This suggested that these genes may play an important and sustained role in lotus petioles formation. However, the few core genes identified both in transcriptional and translational levels may be due to the fact that the genes involved in lotus petioles formation do not express synchronously. As reported by Casas-Vila et al. in *Drosophila melanogaster*, they found unusual behavior of RNA and protein during embryogenesis [32]. This indicates a strong post-translational regulation and the necessity of joint analysis of the transcriptome and proteome in biology.

### 2.6. Functional Enrichment of Quantified DEGs and DEPs

Excepting the 135 core genes, the function of other DEGs and DEPs was analyzed by MapMan software, GO and KEGG annotation tools. The DEPs were classified into 4 clusters according to the fold changes. An abundance of 164 proteins in IFP was 2-fold higher than IVP (Q1), while the abundance of 257 proteins in IFP was 1.5–2-fold higher than IVP (Q2). In addition, the abundance of 346 proteins in IVP was 1.5–2-fold higher than IFP (Q3), while the abundance of 137 proteins in IVP was 2-fold higher than IFP (Q4) (Appendix A). Functional enrichment including GO enrichment, protein domain enrichment, and KEGG pathway enrichment of the DEPs in four clusters was analyzed. Among these 164 proteins in cluster Q1, 13 non-redundant proteins were enriched in cell wall formation and degradation. Among these 137 proteins in cluster Q4, 7 non-redundant proteins were enriched in cell wall biogenesis, cell wall organization and cell wall assembly. In cluster Q4, a total of 9 non-redundant proteins were enriched in polysaccharide including cellulose and hemicellulose biosynthetic process. Nonetheless, in cluster Q2 and Q3, no protein was enriched in cell wall formation and degradation related biologic processes (Figure 6a). This protein functional enrichment analysis coincides with the anatomic results and it indicates that cell wall biosynthesis in IFP and IVP may be different.

Generally, the GO enrichment analysis of these DEGs were shown in (Appendix A). The DEGs were analyzed in upregulated and downregulated gene clusters. It showed that a total of 219, 141 and 87 upregulated genes in IVP were enriched in anatomic structure development, anatomic structure morphogenesis and cell growth. The principal point is that a total of 92 upregulated genes in IVP were enriched in cell wall organization or biogenesis. However, the upregulated genes in IFP were not enriched in pathways related to cell wall organization or biogenesis (Appendix A). In MapMan analysis, a total of 46 non-redundant DEGs and 63 non-redundant DEPs were identified in cell wall biosynthesis and degradation pathway. Moreover, a total of 10 DEGs and 16 DEPs were identified in lignin biosynthesis. More DEPs than DEGs were identified in cell wall biosynthesis and lignin biosynthesis pathway may indicate proteins contribute more to the difference among samples.

### 2.7. Polysaccharide and Lignin Biosynthesis Pathway Were Highly Activated in IVP Compared to IFP

Plant cell walls are composed of polysaccharides, including cellulose, hemicellulose and pectin, lignin, proteins and minerals. Here numerous genes involved in cellulose, hemicellulose, and pectin biosynthesis pathways were identified with a changed expression both at transcriptional and protein level (Table 1). In the DEGs and DEPs identified in this study, 12 genes (3 transcripts and 9 proteins) involved in cell wall precursor synthesis were identified, and all the 12 genes showed expression increment in IVP. A total of 18 genes (6 transcripts and 12 proteins) involved in cellulose and hemicellulose synthesis were identified, and 16 of them showed expression increment in IVP except two transcripts. Moreover, a total of 26 genes (14 transcripts and 12 proteins) involved in cell wall biosynthesis and modifications were identified, and 15 of them showed expression increment in IVP. Nonetheless, a total of 24 genes (11 transcripts and 13 proteins) involved in cell wall degradation were identified and 13 of them showed expression decline in IVP (Table 1, Figure 6b). It indicated that more genes involved in cell wall precursor synthesis, cellulose and hemicellulose synthesis expressed highly in IVP may play foremost roles in its rigidity formation. Cell wall biosynthesis related genes like cellulose synthases (CESA), UDP-glucose 6-dehydrogenases (UGD), UDP-glucuronic acid decarboxylases (UXS), irregular xylem (IRX), and sucrose synthases (SUSY) showed significant higher expression level in IVP than that in IFP (Appendix A).

In the KEGG pathway enrichment analysis, a total of 114 proteins were enriched in 6 different pathways such as starch and sucrose metabolism, amino sugar and nucleotide sugar metabolism, glycosphingolipid biosynthesis, phenylpropanoid biosynthesis, flavonoid biosynthesis and biosynthesis of amino acids (Table 2). Remarkably, 14 proteins highly expressed in IVP and 29 proteins highly expressed in IFP were enriched in phenylpropanoid biosynthesis. Furthermore, the abundance of 10 enzymes including phenylalanine ammonia-lyases (PAL, 3 PAL proteins appear in DEPs), cinnamate 4-hydroxylase (C4H, 1 C4H protein appears in DEPs), 4-(hydroxy)cinnamoyl CoA ligase (4CL, 1 4CL protein appears in DEPs), cinnamoyl CoA reductase (CCR, 1 CCR protein appears in DEPs), cinnamyl alcohol dehydrogenase/sinapyl alcohol dehydrogenase (CAD/SAD, 1 CAD protein appears in DEPs), caffeic acid/5-hydroxyferulic acid *O*-methyltransferase (COMT, 1 COMT protein appears in DEPs), ferulate 5-hydroxylase (F5H, 1 F5H protein appears in DEPs) and hydroxycinnamoyl CoA: shikimate hydroxycinnamoyltransferase/hydroxycinnamoyl CoA: quinate hydroxycinnamoyltransferase (CST/CQT, 1 CST protein appears in DEPs), which are enzymes activated on the upstream of lignin biosynthesis, were higher in IVP than that in IFP. Peroxidase is considered as the last enzyme to catalyze substrate in lignin biosynthesis. In this study, two of peroxidases were highly expressed in IVP compared to IFP and 25 of them were on the contrary. Moreover, the abundance of beta-glucosidase (BGLU, 3 BGLU proteins appear in DEPs) and aldehyde dehydrogenase (REF1, 1 REF protein appears in DEPs), which consume substrates without lignin output in lignin biosynthesis, were highly expressed in IFP (Figure 7). Among 43 identified proteins exhibiting significant changes in lignin biosynthesis pathway between IFP and IVP, 17 of them were selected for qRT-PCR to check their RNA expression. All the seventeen examined genes showed higher mRNA abundance in IVP than that in IFP except gene NNU_03827-RA. Tendencies of transcription level in genes were coincided well to their translation level excluding gene NNU_03827-RA, NNU_19318-RA and NNU_03385-RA (Figure 7). These results indicated that the genes regulating lignin biosynthesis pathway in petioles are regulated similarly at the transcription and translation level, and more genes in lignin biosynthesis pathway were more regulated in IVP compared to IFP.

To evaluate the consequence of genes in lignin biosynthesis pathway being more regulated at transcription and translation level in IVP, metabolites of lignin biosynthesis pathway were quantified in the two petioles types. As shown in (Figure 7, Appendix A), a total of thirteen detected metabolites were quantified in IFP and IVP. Among these metabolites, nine of them exhibited significant changes in abundance between IFP and IVP. The changes abundance of metabolites between IFP and IVP was not exactly coinciding with genes expression, but the precursor abundance of syringyl lignin monomers and hydroxyphenyl lignin monomer were significantly higher in IVP compared to IFP. Considering the higher lignin content in IVP than IFP (Figure 2), the abundance of syringyl lignin and hydroxyphenyl lignin in IVP could be higher than in IFP, and the abundance of guaiacyl lignin may not change.

## 3. Discussion

The molecular mechanism underlying petiole rigidity formation in lotus is important, but far from being fully clarified. Genome sequencing and analysis promote molecular biology research [33]. High throughput sequencing was commonly used in lotus to analyze rhizome formation, flowering time, SNPs and alternative splicing finding and evolution [34,35]. In the present study, gene expression at RNA and protein levels in two petioles types was studied using RNA sequencing and TMT techniques. Based on the integrative analysis of anatomic, transcriptomic and proteomic data, certain genes involved in polysaccharide and lignin biosynthesis pathway were deduced as potentially playing important role in petioles rigidity formation both in transcription and translation level. However, more work needs to be done to explore key genes involved in lotus petioles rigidity formation in future.

### 3.1. Genes Involved in Lotus Petioles Formation Exhibit No Synchrony at Transcriptional and Translation Level

In this study, approximately 75.36% and 18.19% annotated genes in lotus genome have found their transcripts and protein evidence in petioles (Figure 4a). A total of 93.12% proteins had their respective transcripts evidenced in transcriptome data and this ratio is similar to that in other model plant tissue such as *Arabidopsis* root and rice grain [26,30]. Undetected transcripts and proteins may be due to the fact that genes have different spatiotemporal expression characters or the limitation of RNA and protein extraction as well as the detection techniques. Moreover, a total of 335 proteins from proteomics data had no relevant transcripts in transcriptomics data. It was speculated that these genes may have a brief transcription time while their proteins have relatively longer life span in petioles. Similar to a previous study, our results also pointed out the un-synchronous in transcriptional and translation of genes in lotus [32]. The poor correlation of total identified transcripts and proteins suggest that a proportion of transcripts may not be translated into proteins and the higher correlation of DEGs and their corresponding DEPs suggest these genes have similar function in petioles development (Figure 5a,d). Inconsistency of genes in mRNA and protein level was similar with other previous studies [30,31,32]. Furthermore, it suggests that the transcriptomics and proteomics analysis are strategically complementary and have equal importance in lotus petioles development analysis [28,29].

### 3.2. Anatomic Evaluation of Lotus Petioles Indicated Cell Wall Thickness and Cell Differentiation Affect Petiole Rigidity

In spite of some anatomic evaluations being presented on Sacred lotus, no research has specifically explored the difference of the two types of petioles. In this present study, we found that IFP and IVP have similar anatomic structure. However, there are some differences between IFP and IVP. Usually, IVP had more collenchyma tissue, more vascular bundles and thicker xylem vessel cell wall (Figure 2), which enable the plant has more strength to keep its architecture [36]. Genes involved in cell wall polysaccharides biosynthesis may play key roles in its anatomic structure formation. In this study, 5 cellulose synthases, which plays important roles in cell wall formation [37,38], showed higher abundance in IVP than that in IFP. Specifically, there were more layers of collenchyma cell in IVP than that in IFP to support petioles stand upright. The vascular bundles and the thicker xylem vessel cell wall are the usual structure in stem or petioles of *Arabidopsis* [39], rice [40], Populus [41,42], as well as bamboo [43] to provide mechanical support. The more abundance of these structures in IVP indicate that more mechanical strength is generated. Additionally, the different amount of cellulose and lignin and the diverse structure of the cell wall of IVP and IFP may suggest that the cell wall structure could be the main reason for different petioles observed in lotus.

### 3.3. Abundance of Lignin Biosynthesis Related Proteins Significantly Higher in IVP Compared to IFP Supports Contribution of Lignin Biosynthesis in Petiole Rigidity

It has been established that a constant energy supply is a prerequisite for plant growth. Therefore, high carbohydrate metabolism in cell is necessary for growth of petioles. Despite the rapid growth, the floating leaf petiole was phenotypically weak, indicating less deposition of cell wall strengthening compounds and biologic process enrichment gives more clues for cell wall biosynthesis or organization related proteins, demonstrating that they are highly expressed in IVP.

Most importantly, after KEGG pathway enrichment analysis, numerous proteins related to cell wall polysaccharides and lignin biosynthesis were identified from 904 significantly differentially expressed proteins (Figure 6, Appendix A). Phenylpropanoid and other flavonoids were reported to respond to both abiotic and biotic stimuli [44]. Additionally, flavonoids biosynthesis and lignin biosynthesis were reported have some connections in plant [45,46]. Lignin gets deposited in the matrix of cellulose micro fibril, which strengthens the cell wall. This study showed the significantly enriched phenylpropanoid biosynthesis/lignin biosynthesis pathway and identified major enzymes involved in lignin biosynthesis (Figure 7, Table 2). Furthermore, 10 abundant enzymes that trigger lignin biosynthesis at the beginning until the formation of lignin monomer were higher in IVP than IFP, which indicates that IVP had more effective lignin deposition compared to IFP.

The transcription level of the selected enzymes genes in lignin biosynthesis pathway were checked in petioles. The results suggested that the genes involved in the regulation of lignin biosynthesis pathway appear at both transcription and translation levels exhibiting similar tendencies in both. Consequently, analyses of metabolites in lignin biosynthesis imply that more lignin was synthesized in IVP compared to IFP (Figure 2 and Figure 7).

### 3.4. Abundance of Cell Wall Polysaccharides Biosynthesis Related Proteins Significantly Higher in IVP Than That in IFP

Cell wall polysaccharides biosynthesis is more complicated than lignin biosynthesis. Biologic processes relevant to cellulose and hemicellulose biosynthesis affect cell wall polysaccharides biosynthesis [47,48]. Pathways such as starch and sucrose metabolism, amino sugar and nucleotide sugar metabolism, and biosynthesis of amino acids were significantly enriched in this study (Figure 7, Table 2). Cellulose is the most abundant biopolymer deposited on the plant cell wall. Approximately 40–50% cellulose stored in secondary cell wall of wood tissue, which mainly contributes to the bodily structure [47]. Studies in *Arabidopsis*, Populus and other plants suggest that various genes involved in cellulose biosynthesis play fundamental roles in cell wall formation and thickening. A total of 10 cellulose synthases [49] mediate cellulose synthesis in both primary [50] and secondary walls in *Arabidopsis* [51,52]. Previous researches showed that some cellulose synthases in Populus participate in cell wall formation [25,53,54] and other genes including KORRIGAN1 (KOR1), sucrose synthase gene and transcription factors are also involved [42,55,56,57]. Specifically, the abundance of four CESAs (cellulose synthases) including NNU_09561-RA, NNU_12044-RA, NNU_21632-RA and NNU_01080-RA were significantly expressed higher in IVP compared to IFP, suggesting that more cellulose is deposited in IVP (Figure 2 and Figure 3). Similarly, abundance of three UGD (UDP-glucose 6-dehydrogenase) and one SUSY (sucrose synthase), for instance, NNU_03659-RA, NNU_07386-RA, NNU_04520-RA and NNU_19077-RA were significantly enriched higher in IVP. These proteins have similar functions in cellulose biosynthesis as their homologous proteins in *Arabidopsis* and Populus [58,59]. Furthermore, hemicellulose biosynthesis related proteins such as HEX (hexokinase), UXS (UDP-glucuronic acid decarboxylase), IRXs (irregular xylem), GATL (galacturonosyltransferase) and UXT (UDP-xylose transporter) have been reported to be involved in cell wall hemicellulose biosynthesis [39,60,61,62]. In our study, several UXS (NNU_15122-RA, NNU_12302-RA, NNU_10172-RA and NNU_25253-RA), IRX (NNU_20069-RA, NNU_12476-RA, NNU_13626-RA, NNU_13619-RA, NNU_22319-RA, NNU_03759-RA, NNU_18746-RA and NNU_11544-RA) and UXT (NNU_12576-RA, NNU_24078-RA) were identified and most of them showed significantly higher abundance in IVP than in IFP, suggesting the huge contribution of hemicellulose biosynthesis in cell wall formation. The function of these proteins involved in polysaccharides biosynthesis need more experiments in vivo to confirm because the presence of glycoside hydrolases does not mean that the polysaccharides are degraded [63]. Their expression of these genes at mRNA and protein level may have more complicated relation with polysaccharides deposition in cell wall.

## 4. Materials and Methods

### 4.1. Plant Growth and Petiole Collection

The sacred lotus cultivar “Ancient Chinese lotus” was cultivated in Wuhan Botanical Garden, Chinese Academy of Sciences (N30°32′44.02″, E114°24′52.18″). There are two ecotypes of N. nucifera: temperate lotus and tropical lotus (N19°~N43° in China) [64]. “Ancient Chinese lotus” is temperate lotus. Its seed starts to germinate in April, flower blooms from June to August, leaf withers in September and October and dormant buds remains from November to March of the following year with an enlarged rhizome [65]. In middle May, the dormant buds sprout underground. In terms of leaf development, it could be divided into two stages. At floating leaf development stage, the newly developed leaf is the floating leaf, which is folded until emerging above the water surface. As the first floating leaf grows into maturity, the second leaf appears. After about several floating leaves have emergence, the succeeding leaves are vertical leaves. At the vertical leaf development stage, the newly developed vertical leaf is folded, which will grow bigger and unfold when the petiole is long enough to support leaf far above the water surface. Four kinds of petioles samples were collected at different stages. At the floating leaf development stage, initial floating leaves’ petiole (IFP) was collected from floating leaf when it just floats on water surface. Mature floating leaves’ petiole (MFP) was collected from floating leaf when its shape and size had stopped changing. At the vertical leaf development stage, initial vertical leaves’ petiole (IVP) was collected when the folded leaf just emerges out of water surface. Mature vertical leaves’ petiole (MVP) was collected when its shape and size stopped changing (Appendix A). In this study, rhizome of “Ancient Chinese lotus” was planted in concrete container (length, 2 m; width,1 m; depth, 80 cm) with loam soil is about 50 cm in depth on 5 April 2016. The containers were constructed outside under natural light conditions. Fifty grams of fertilizer (SAN AN; STANLEY, Shandong, China) was applied to container every 2 weeks during the growing season. In field, lotus usually was cultured in farmland or pond with different depth of water (20 cm to 100 cm). The two different types of petioles were always observed in different depth of water in filed. In order to exclude the effect of water depth on ability of leaf stand erect on water, lotus was cultivated in low water depth. this means the length of two types of petioles was longer than water depth.

### 4.2. Morphologic Observation

#### 4.2.1. Petiole Microscopic Observation

Petiole segments (the middle section of the petiole) were cut and fixed in FAA (formaldehyde–acetic acid–alcohol solution) solution (5% glacial acetic acid, 5% formaldehyde and 70% ethanol) at room temperature for 24 h. The segments were then rinsed with tap water and dehydrated in ethanol baths ranges of 70% ethanol to 100% ethanol. After dehydration, ethanol was replaced by chloroform progressively. Then finely ground paraffin was added into chloroform and incubated it at 36 °C for 2 h. The small segments were transferred into xylene solution containing 50% and 75% paraffin and incubated at 42 °C and 50 °C for 2 h successively. After that, the small segments were transfer into pure paraffin solution at 58 °C for 1 h and this process was repeated three times. Finally, the sample were embedded using pure paraffin. Samples were cut with a rotary microtome Leica RM2265 (Leica, Benshein, Germany). Sections (10 μm) were stained using fast green and counterstained using safranin solution (Biosharp, Hefei, China) and observed under an optical microscope (Olympus BX53, Tokyo, Japan). Safranin appears red in lignified, suberized or cutinized cell walls. Fast Green (Biosharp, Hefei, China) presents green in cytoplasm and cellulosic cell walls. Petiole segments from three independent plants were collected, and at least 3 sections per plant were employed for microscopy observation.

#### 4.2.2. Breaking Resistance Measurement

Petioles (petiole length was about 30 cm) with same diameter (5 mm) from IFP and IVP were cut into equally long stem segment (20 cm). The breaking resistance of the middle point of petiole was measured using a prostrate tester (DIK 7400, daiki rika kogyo co. ltd., Tokyo, Japan). The distance between tester fulcra was set at 10 cm (Figure 1g,h). The petiole was pressed using prostrate tester until petiole breaking point and the test readings were recorded. Breaking resistance was represented by the manual force added on petiole. Breaking resistance (F) = (test reading × 39.2 N) ÷ 40 [66,67]. Six biologic petioles of each type of petiole were tested in this experiment.

#### 4.2.3. Transmission Electron Microscope

Petiole segment (the middle section of the petiole) was infiltrated in 4% (*v/v*) glutaraldehyde in phosphate buffered solution at 4 °C overnight (PBS, 33-mM Na_2_HPO_4_, 1.8-mM NaH_2_PO_4_ and 140-mM NaCl [pH 7.2]). Then the petiole segment was fixed in 1% (*w/v*) osmium tetroxide (Biosharp, Hefei, China) and dehydrated through a gradient of ethanol and eventually embedded in Spurr’s resin (Biosharp, Hefei, China). Petiole sections (100 nm) were stained using uranyl acetate and lead citrate. Lastly, the petiole sections from three independent petioles from three biologic lotus were visualized using HT7700 transmission electron microscope (HITACHI, Tokyo, Japan) and the cell-wall thickness was measured from ten cells on each section using Hitachi TEM system microscopy software (Gatan, Pleasanton, CA, USA).

#### 4.2.4. Determination of Crude Cellulose and Lignin

Crude cellulose was measured by acid digestion method using cellulose detective kit (Sinobestbio, Shanghai, China). The weighed samples (the middle section of the petiole) were grinded with 80% ethanol and washed with ethanol and acetone. After the collected residue was digested by concentrated sulfuric acid and the supernatant was used to detect cellulose content. Lignin content was measured according to Klason lignin methods [68]. Six biologic petioles of each type of petiole were tested in this experiment.

### 4.3. RNA Isolation and Illumina Sequencing

#### 4.3.1. Total RNA Extraction

Total RNA was extracted from about 0.5 g of samples with TRIzol reagent (Invitrogen, Carlsbad, CA, USA) according to the standard protocol. RNA quality and concentration were evaluated using the 6000 Pico LabChip of the Agilent 2100 Bioanalyzer (Agilent, Santa Clara, CA, USA) and a NanoDrop spectrophotometer (Thermo Scientific, Waltham, MA, USA), respectively.

#### 4.3.2. cDNA Synthesis and Illumina Sequencing

The cDNA was synthesized by cDNA synthesis kit (Bio-Rad, Hercules, CA, USA) according to the manufacturer’s instructions. After purification, fragments were enriched by PCR amplification for cDNA library construction. Then, library products were sequenced with Illumina HiSeq^TM^ 2000 platform (San Diego, CA, USA). Each sample had three biologic replicates from three individual plants. The raw data of this project was deposited in NCBI (www.ncbi.nlm.nih.gov/bioproject) and can be downloaded with identifier PRJNA642670.

#### 4.3.3. Bioinformatics Analysis

RNA sequencing data processing, gene expression calculation, differentially expressed genes (DEGs) identification, gene ontology (GO) enrichment analysis and pathway enrichment were performed in BMK Cloud (Beijing, China, http://www.biocloud.net/). A false discovery rate (FDR) <1% and an absolute value of log2Ratio >1 was set as the threshold to judge the significance of gene expression difference between two petioles types. Generally, the raw RNA sequencing data were cleaned by removing adapter sequences, reads containing ploy-N and filtered with low-quality sequences (Q < 20) from raw data. All the downstream analyses were based on clean data with high quality. Clean reads were aligned to the reference genome sequence using the algorithm Tophat2 [69]. Tolerance parameters in Tophat2 were set as default with no more than two bases mismatches. Then gene function was annotated based on the following databases: Nr (NCBI non-redundant protein sequences); Nt (NCBI non-redundant nucleotide sequences); Pfam (Protein family); KOG/COG (Clusters of Orthologous Groups of proteins);Swiss-Prot (A manually annotated and reviewed protein sequence database); KO (KEGG Ortholog database); GO (Gene Ontology). Gene expression levels were estimated by FPKM (fragments per kilobase of transcript per million fragments mapped). Differential expression analysis of two groups was performed using the DESeq R package (1.10.1). DESeq provide statistical routines for determining differential expression in digital gene expression data using a model based on the negative binomial distribution. The resulting *p*-values were adjusted using the Benjamini and Hochberg’s approach for controlling the false discovery rate.

### 4.4. Proteomics Analysis

#### 4.4.1. Protein Extraction and Trypsin Digestion

Petioles were ground and transferred to a 5-mL centrifuge tube containing homogenizing buffer (8-M urea, 2-mM ethylenediaminetetraacetic acid (EDTA), 10-mM dl-dithiothreitol (DTT) and 1% protease inhibitor cocktail). The protein was precipitated using cold TCA-acetone solution for 2 h at −20 °C. Then, the remaining precipitate was washed with cold acetone. The protein was re-dissolved in lysis buffer (8-M urea, 100-mM triethylammonium bicarbonate (TEAB), pH 8.0) and the protein concentration was measured by BCA methods [70]. The protein solution was reduced by 10-mM DTT for 30 min at 37 °C and alkylated with 20-mM iodoacetamide (IAM) at 37 °C in darkness for 45 min. Trypsin was added with a 1:50 trypsin-to-protein mass ratio for overnight digestion. All regents used in this section were supplied by Sangon Biotech Company (Shanghai, China). Six biologic petioles of each type of petiole were used in this experiment.

#### 4.4.2. TMT Labeling

After trypsin digestion, the peptide was desalted using ZipTip C18 column (Millipore, Bedford, MA, USA). Peptide was dissolved in 0.5-M TEAB and processed according to the manufacturer’s protocol for 6-plex TMT kit (Thermo Fisher Scientific, Waltham, MA, USA). The solvable peptides were desalted using a MonoSpin C18 column (GL 217 Sciences, Tokyo, Japan) [70].

#### 4.4.3. Quantitative Proteomic Analysis by LC-MS/MS

Peptides in each sample were separated into 18 fractions and dried by vacuum centrifugation (Labconco, Kansas, MO, USA). Then peptides were dissolved in 0.1% formic acid (FA) and loaded directly onto the reversed-phase pre-column (Acclaim PepMap 100, Thermo Scientific, Waltham, MA, USA). The peptides were separated by the reverse phase analysis column (Acclaim PepMap RSLC, Thermo Scientific, Waltham, MA, USA) according to the following conditions. The 6% to 22% solvent B (0.1% FA in 98% ACN, acetonitrile) for 22 min, 22% to 36% for 10 min, escalating to 85% in 5 min and held at 85% for the last 3 min. Flow rate was held at 300 nL/min on an EASY-nLC 1000 ultra-performance liquid chromatography system (UPLC)(Thermo Scientific, Waltham, MA, USA). Peptides were detected in the Orbitrap (Thermo Scientific, Waltham, MA, USA) at a resolution of 70,000 and instrument parameters were set as follows: Normalized collision energy (NCE) value was 30; ion fragments resolution was 17,500. The electrospray voltage was 2.0 kV. The *m/z* scan range is 350 to 1800. Fixed first mass is 100 *m/z* [70]. A data-dependent analysis method that alternated between primary MS scan followed by 20 MS/MS scans was applied for the top 20 precursor ions above a threshold ion count of 1E4 in the MS survey scan with 30.0 s dynamic exclusion.

#### 4.4.4. Database Searching

The acquired MS/MS data were processed using Proteome Discoverer (version 2.1.0.81, Thermo Scientific, Waltham, MA, USA). Tandem mass spectra were searched against the sacred lotus genome database with mascot search engine (https://bioinformatics.psb.ugent.be/plaza/versions/plaza_v4_dicots/). The parameters were set as follows: Trypsin/P was chosen as enzyme and the maximum missing cleavages was 2. The number of modification and charge in each peptide were set up to 5. Mass error tolerance was set to 10 ppm and 0.02 Da for precursor and fragment ions, respectively. Fixed modification included carbamidomethylation on Cys and TMT-6-plex on Lys and N-term. Variable modification contained oxidation on Met and FDR (false discovery rate) threshold value 1% was applied in protein, peptide and modification site identification. Minimum peptide length was set at 7.

#### 4.4.5. Protein Annotation, Functional Classification and Enrichment

Proteins were classified by GO annotation method. GO annotation proteome was derived from the UniProt-GOA database (http://www.ebi.ac.uk/GOA/) [71,72]. KEGG database was used in the identification of enriched pathways [71,72]. A two-tailed Fisher’s exact test was employed to test the enrichment of DEPs against all identified proteins. The GO or pathway with a corrected *p*-value < 0.05 was considered significant.

### 4.5. Metabolism Analysis

#### 4.5.1. Metabolite Extraction

Freeze-dried petiole was crushed using a mixer mill (MM 400, Retsch, Germany). One hundred milligrams of powder were weighed and stirred in 70% methanol (1.0 mL). After centrifugation, the supernatant containing extracts were absorbed (CNWBOND Carbon–GCB SPE Cartridge, 250 mg, 3 mL, Shanghai, China and filtrated before LC-MS analysis. Six biologic petioles of each type of petiole were tested in this experiment.

#### 4.5.2. Metabolite Analysis

Metabolite profiling was carried out using a widely targeted metabolome method by Wuhan Metware Biotechnology Co., Ltd. (Wuhan, China, http://www.metware.cn/). The sample extracts were analyzed with LC-ESI-MS/MS system. The analytical conditions were as follows. Column (Waters, 1.8 µm, 2.1 mm × 100 mm) and buffer A (water containing 0.04% acetic acid) and buffer B (acetonitrile containing 0.04% acetic acid) were used. The gradient program in HPLC system (SCIEX, Redwood City, CA, USA) was set as follows: At the first 12 min, mobile phase containing buffer A 5% and buffer B 95%. In the next three min, mobile phase was changed into buffer A 95% and buffer B 5%. The flow rate was set as 0.40-mL/min and temperature was set at 40 °C. A total of 5-μL sample was injected in each run. The eluate was alternatively connected to MS (API 4500 Q TRAP LC/MS/MS System, SCIEX, Redwood City, CA, USA).

### 4.6. Validation of Transcriptomic and Proteomic Data Using qRT-PCR and PRM Analysis

#### 4.6.1. qRT–PCR

Total RNA was extracted by TRIzol reagent (Invitrogen, Carlsbad, CA, USA) and 2 µg total RNA was used for reverse transcription using Rever Tra Ace-α-First Strand cDNA synthesis Kit (TOYOBO, Osaka, Japan). qRT-PCR was performed on CFX96 Real Time System (BIO-RAD, Hercules, CA, USA) with SYBR green fluorescence. The data were normalized based on relative transcript level of actin (NNU_24864). Three biologic petioles of each type of petiole were tested in this experiment.

#### 4.6.2. Parallel Reaction Monitoring (PRM)

The digested peptides were dissolved in 0.1% formic acid and directly loaded onto a reversed-phase analytical column (15 cm length, 75 μm i.d.). The parameters of UPLC (Thermo Scientific, Waltham, MA, USA) were set as mentioned above. The parameters of MS were set as follows: The electrospray voltage was 2.0 kV. The *m/z* scan range was 350 to 1000 for full scan and the resolution was 35,000. Peptides were then selected for MS/MS using NCE setting at 27 and the resolution of fragments was 17,500. The maximum IT was set at 20 milliseconds for full MS and auto for MS/MS. The isolation window for MS/MS was set at 2.0 *m/z*. The acquired MS data were identified and quantified using Skyline software (v.3.6) [73]. MS proteomics data in this study were deposited in the ProteomeXchange Consortium (http://proteomecentral.proteomexchange.org) via the PRIDE partner repository [74]. Data are available via ProteomeXchange with identifier PXD009136.

### 4.7. Statistical Analysis

The data were analyzed using the Student’s *t*-test to evaluate the statistical significance to compare the two groups, and one-way ANOVA test to compare multiple groups (*p* < 0.05).

## 5. Conclusions

In lotus growth, the floating leaves and erect rigid leaves develop subsequently. These two petioles types appear like lateral branches growing during its growing season. Due to the fact that floating leaf and vertical leaf experience similar ambient, the differentiation of petioles rigidity in lotus is not affected by environmental changes. To investigate the mechanism controlling the rigidity of lotus petiole, we performed integrated omics analysis in this study. Anatomically, more vascular bundles, collenchyma and thicker cell wall were observed in IVP than that in IFP. Many genes involved in cell wall biosynthesis and lignin biosynthesis exhibited differentially expression at both mRNA and protein levels. Gene function and pathway enrichment analysis displayed that DEGs and DEPs were significantly enriched in cell wall biosynthesis and lignin biosynthesis. Interestingly, the bulk of identified DEPs in lignin biosynthesis were up regulated in IVP, suggesting that the differences in lignin biosynthesis in the two types of lotus petioles may be responsible for the observed different rigidity. Despite our results are still far from underlying the rigidity of lotus construction, these findings afford novel clues for better understanding of the gene network involved in lotus petioles formation. In addition, roles of decisive candidate gene involved in cell wall and lignin biosynthesis in lotus should be accurately interpreted in future studies.

## Figures and Tables

**Figure 1 ijms-21-05087-f001:**
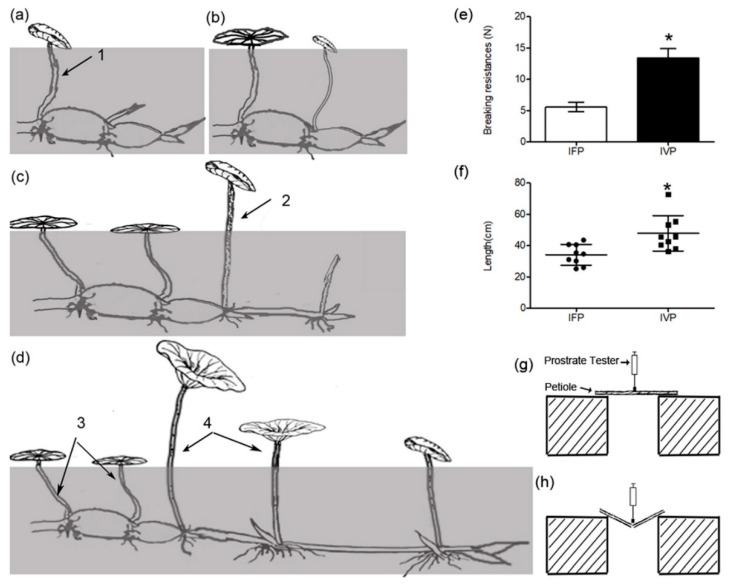
Schematic diagram of growing processes of floating leaf and vertical leaf. (**a**–**d**) stand different growth stages of leaves. Arrow 1 indicates initial floating leaves’ petiole (IFP), arrow 2 indicates initial vertical leaves’ petiole (IVP), arrow 3 indicates mature floating leaves’ petiole (MFP) and arrow 4 indicates mature vertical leaves’ petiole (MVP), respectively; (**e**) measurements of breaking resistances in two type petioles were performed using a prostrate tester; (**g**,**h**) show status of petioles before and after putting pressure on it; (**f**) shows the length of petioles which were cultured in pot with 20 cm depth water. Data are means ± SD from 9 independent petioles (IFP and IVP). Asterisks indicate significant changes according to Student’s t-test (* *p* < 0.05).

**Figure 2 ijms-21-05087-f002:**
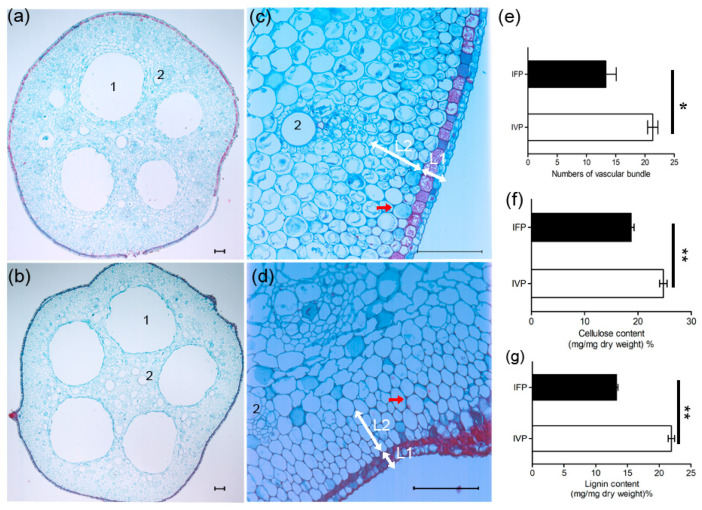
Transverse sections of IFP and IVP stained with fast green and the counterstain safranin. (**a**,**c**) and (**b**,**d**) show transverse sections of IFP and IVP, respectively. Arabic numerals 1 and 2 indicate big air cavities and xylem vessel. L1 and L2 indicate the epidermal and cortex. Red arrow shows collenchyma. Bars in figures indicate 100 μm; (**e**) statistics of vascular bundles number in IFP and IVP, data are means ± SD from 10 independent transverse sections (IFP and IVP); (**f**) determination of crude cellulose and (**g**) lignin. Data are means ± SD from 3 independent biologic repeats. Asterisks and double asterisks indicate significant changes compared to the control as assessed according to Student’s *t*-test (* *p* < 0.05 and ** *p* < 0.01).

**Figure 3 ijms-21-05087-f003:**
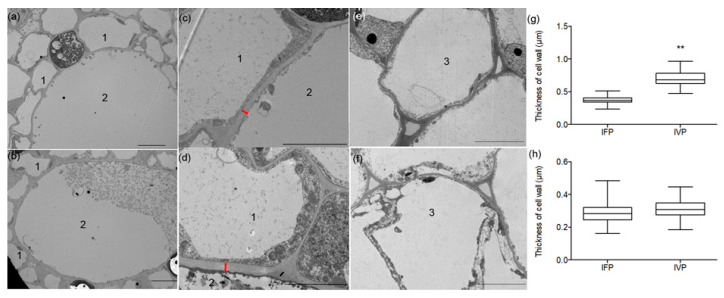
Thickness of cell wall in lotus petioles was observed using a transmission electron microscope. (**a**,**c**) shows cell wall thickness of xylem vessels in IFP; (**b**,**d**) shows cell wall thickness of xylem vessels in IVP; (**e**,**f**) show thickness of parenchymatous cell wall of IFP and IVP, respectively. The Arabic numerals 1 indicates sclerenchyma cell; Arabic numerals 2 indicates xylem cell, The Arabic numerals 3 indicates parenchymatous cell; (**g**,**h**) statistics of thickness of the cell wall of xylem vessels and parenchymatous cell in IFP and IVP, respectively. Data are means ± SD from 10 independent transverse sections (IFP and IVP). Double asterisks indicate significant changes compared to IFP as assessed according to Student’s *t*-test (** *p* < 0.01). Bars in figures indicate 10 μm.

**Figure 4 ijms-21-05087-f004:**
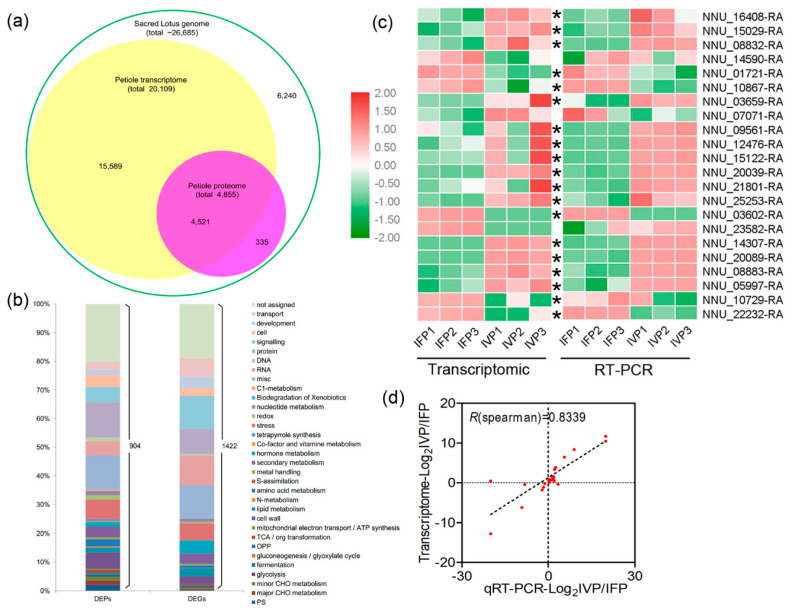
Comparison of protein and transcript abundance in lotus petioles. (**a**) Congruency between the detected transcripts and proteins of lotus petioles; (**b**) functional classification and distribution of differentially expressed genes (DEGs) and differentially expressed proteins (DEPs); (**c**) heatmap of 22 selected genes performed by transcriptomic and RT-PCR data. Asterisk indicates genes have a similar expression pattern in transcriptomic and RT-PCR experiments; (**d**) correlations of gene expression between transcriptomic and RT-PCR data. R represents the Pearson correlations coefficient.

**Figure 5 ijms-21-05087-f005:**
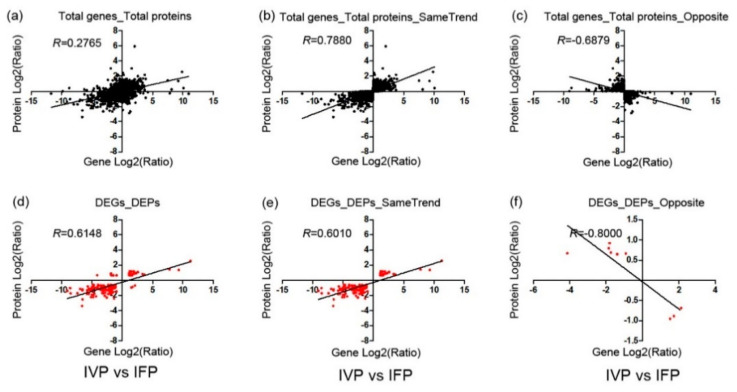
Concordance between changes in the abundance of mRNA and its encoded protein in IFP and IVP. R, Pearson correlations coefficient of the comparisons between fold changes of proteins and transcripts. (**a**) the expression levels of all the transcripts and their corresponding quantified proteins from IVP vs. IFP showed weak correlation (r = 0.2765). (**d**) a stronger correlation was discerned between the DEGs and their corresponding DEPs (r = 0.6148). The expression ratio of proteins and their corresponding mRNAs with the same or opposite trend (both upregulated or both downregulated) were also plotted, and higher positive or negative correlation was indicated (**b**,**c**,**e**,**f**).

**Figure 6 ijms-21-05087-f006:**
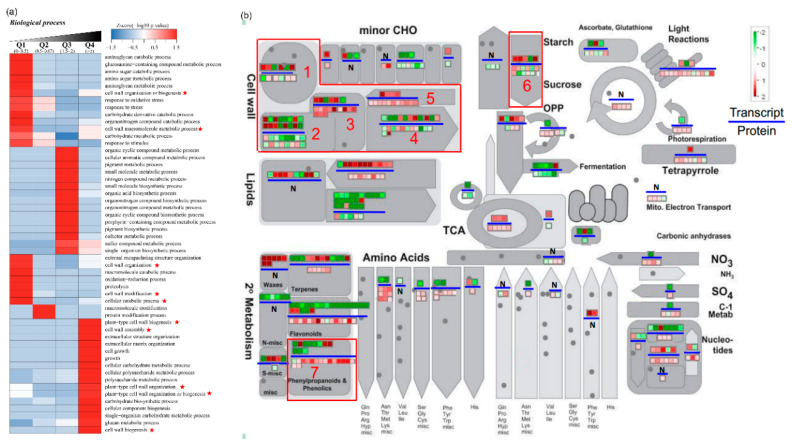
GO and MapMan analysis the function of DEGs and DEPs in lotus. (**a**) Function enrichment analysis of proteins with significant changes in abundance between IFP and IVP. Abundance of 164 proteins in IFP was 2-fold higher than IVP (Q1) and abundance of 257 proteins in IFP was 1.5–2-fold higher than IVP (Q2). Also, abundance of 346 proteins in IVP was 1.5–2-fold higher than IFP (Q3) and abundance of 137 proteins in IVP was 2-fold higher than IFP (Q4). Red star indicates cell wall related process; (**b**) MapMan analysis shows that DEGs and DEPs in different metabolic pathways in lotus petioles. Red squares indicate enhancement and green squares were the opposite. Squares above the blue line represent the transcripts and squares below the blue line represent the proteins.N means not found. Arabic numerals 1 to 7 in red frame represent cell wall proteins, pectin esterases, cellulose synthesis, cell wall degradation, cell wall precursor synthesis, major CHO metabolism and lignin biosynthesis pathways.

**Figure 7 ijms-21-05087-f007:**
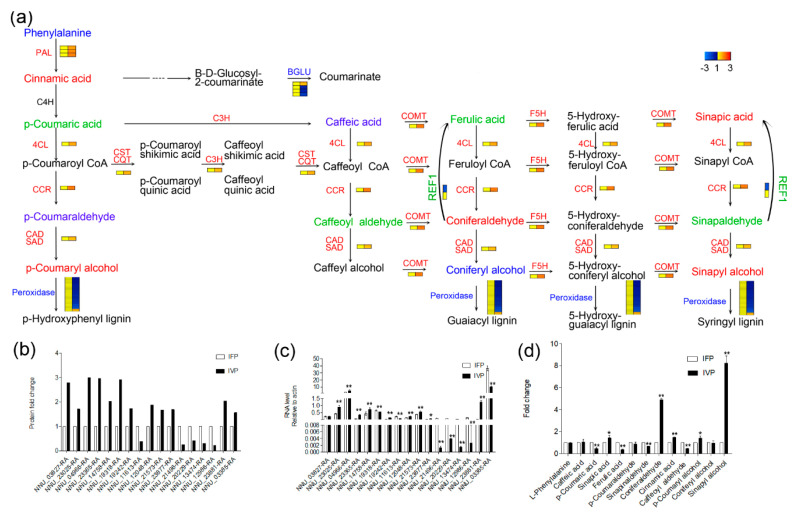
Overview of lignin biosynthesis pathway on transcription level, translation level and metabolite level. (**a**) Substrate, product and enzymes in whole pathway are written in red, green, blue and black color. Red color means abundance of substrate, product or enzymes were higher in IVP than IFP, green color means abundance of substrate, product or enzymes were higher in IFP than IVP and blue color means abundance of substrate and product were similar in two petioles or enzymes which catalyze same action show opposite abundance between IVP and IFP. Color bar shows changes in abundance of enzymes which were quantified in proteomics; (**b**,**c**) abundance of selected enzymes in lignin biosynthesis pathway and mRNA expression level of genes encoding selected enzymes in lignin biosynthesis pathway; (**d**) abundance of metabolites in lignin biosynthesis pathway. Data are means ± SD from 3 independent biologic repeats. Asterisks and double asterisks indicate significant changes compared to control as assessed according to Student’s *t*-test (* *p* < 0.05 and ** *p* < 0.01).

**Table 1 ijms-21-05087-t001:** DEPs and DEGs involved in cell wall biosynthesis.

ID	Name	Log2(IVP/IFP)	Log2(IVP/IFP)
**cell wall precursor synthesis**			
NNU_03659-RA	UDP-glucose 6-dehydrogenase 4-like	0.9	Null
NNU_04520-RA	UDP-glucose 6-dehydrogenase 1	0.7	Null
NNU_07386-RA	UDP-glucose 6-dehydrogenase 1-like	0.9	Null
^a^ NNU_08832-RA	probable rhamnose biosynthetic enzyme 1	1.2	1.3
NNU_10172-RA	probable rhamnose biosynthetic enzyme 1	0.8	Null
NNU_12302-RA	UDP-glucuronic acid decarboxylase 6	0.8	Null
NNU_15122-RA	UDP-glucuronic acid decarboxylase 6	1.2	Null
NNU_25253-RA	hypothetical protein OsI_05369	0.6	Null
NNU_26559-RA	probable rhamnose biosynthetic enzyme 1	1.0	Null
NNU_00650-RA	probable arabinose 5-phosphate isomerase	Null	1.4
NNU_21054-RA	bifunctional UDP-glucose 4-epimerase and UDP-xylose 4-epimerase 1	Null	1.9
**cell wall proteins**			
NNU_01080-RA	glucomannan 4-beta-mannosyltransferase 2-like	0.9	Null
NNU_25605-RA	fasciclin-like arabinogalactan protein 17	0.7	Null
^a^ NNU_11213-RA	fasciclin-like arabinogalactan protein 13	−1.6	−3.2
NNU_12269-RA	fasciclin-like arabinogalactan protein 4	Null	1.4
NNU_15965-RA	fasciclin-like arabinogalactan protein 7	Null	1.8
NNU_06301-RA	leucine-rich repeat extensin-like protein 6	Null	−8.3
NNU_16861-RA	leucine-rich repeat extensin-like protein 4	Null	3.0
NNU_24457-RA	leucine-rich repeat extensin-like protein 4	Null	3.0
NNU_25277-RA	glucomannan 4-beta-mannosyltransferase 9	Null	1.7
**cell wall degradation**			
NNU_05055-RA	probable polygalacturonase	0.9	Null
NNU_11467-RA	hypothetical protein PHAVU_009G016100 g	2.2	Null
NNU_11761-RA	probable polygalacturonase	1.3	Null
NNU_23253-RA	probable pectate lyase 18	0.7	Null
NNU_23813-RA	GDSL esterase/lipase At5g14450 isoform X3	0.9	Null
NNU_00300-RA	probable polygalacturonase isoform X1	−0.7	Null
NNU_05224-RA	probable rhamnogalacturonate lyase B isoform X1	−0.8	Null
^a^ NNU_10867-RA	alpha-L-arabinofuranosidase 1-like	−1.2	−2.1
NNU_11580-RA	polygalacturonase inhibitor-like	−0.9	Null
NNU_13918-RA	putative beta-D-xylosidase	−0.9	Null
NNU_22026-RA	probable pectate lyase 18	−2.9	Null
NNU_07943-RA	probable polygalacturonase	Null	−2.3
NNU_11529-RA	probable polygalacturonase isoform X2	Null	−1.9
NNU_11581-RA	polygalacturonase inhibitor-like	Null	−9.0
NNU_18600-RA	lysosomal beta glucosidase-like isoform X4	Null	−1.0
NNU_19090-RA	polygalacturonase inhibitor-like	Null	−2.8
NNU_13976-RA	probable polygalacturonase	Null	1.3
NNU_23205-RA	probable polygalacturonase isoform X1	Null	1.3
NNU_23792-RA	probable polygalacturonase non-catalytic subunit JP650	Null	1.8
NNU_26608-RA	alpha-L-fucosidase 1-like	Null	1.1
**cell wall modification**			
^a^ NNU_15029-RA	xyloglucan endotransglucosylase/hydrolase protein 22-like	1.1	1.9
^a^ NNU_16408-RA	probable xyloglucan endotransglucosylase/hydrolase protein 8	0.9	1.6
NNU_17562-RA	expansin-A13-like	0.7	Null
NNU_25629-RA	probable xyloglucan endotransglucosylase/hydrolase protein 6	1.0	Null
NNU_12232-RA	expansin-A8-like	−1.1	Null
NNU_12958-RA	expansin-A4	−0.8	Null
NNU_24404-RA	expansin-A8-like	−1.1	Null
NNU_24832-RA	probable xyloglucan endotransglucosylase/hydrolase protein 23	−0.8	Null
NNU_20658-RA	pectinesterase-like	0.9	Null
^a^ NNU_01721-RA	probable pectinesterase/pectinesterase inhibitor 51	−1.1	−1.6
NNU_05006-RA	pectinesterase	−1.9	Null
NNU_05086-RA	pectinesterase	−1.9	Null
NNU_08272-RA	protein notum homolog	−1.1	Null
NNU_11705-RA	pectinesterase-like	−1.5	Null
NNU_15192-RA	pectinesterase-like	−0.8	Null
NNU_18238-RA	pectinesterase 2-like	−0.7	Null
NNU_05158-RA	putative expansin-A17 isoform X1	Null	−8.6
NNU_05160-RA	putative expansin-A17	Null	−4.6
NNU_23652-RA	expansin-A15-like	Null	−2.9
NNU_15361-RA	brassinosteroid-regulated protein BRU1-like	Null	1.6
NNU_16495-RA	xyloglucan endotransglucosylase/hydrolase protein 9	Null	2.3
NNU_24956-RA	probable xyloglucan endotransglucosylase/hydrolase protein 33	Null	4.4
NNU_12324-RA	probable pectinesterase/pectinesterase inhibitor 41	Null	−2.6
NNU_14002-RA	pectinesterase-like	Null	−1.7
NNU_14557-RA	pectinesterase 2-like	Null	−5.6
NNU_24388-RA	probable pectinesterase/pectinesterase inhibitor 41	Null	−1.4
NNU_05007-RA	pectinesterase/pectinesterase inhibitor PPE8B	Null	2.3
NNU_18245-RA	probable pectinesterase/pectinesterase inhibitor 41	Null	3.3
NNU_18519-RA	L-ascorbate oxidase homolog	Null	1.7
**cellulose synthase**			
NNU_09561-RA	probable cellulose synthase A catalytic subunit 5	2.4	Null
NNU_12044-RA	cellulose synthase A catalytic subunit 7	1.3	Null
NNU_21632-RA	probable cellulose synthase A catalytic subunit 1	1.2	Null
NNU_07451-RA	protein COBRA-like isoform X1	1.4	Null
NNU_07455-RA	COBRA-like protein 4	1.3	Null
NNU_13962-RA	COBRA-like protein 7	1.3	Null
NNU_21801-RA	endoglucanase 9-like	1.1	Null
NNU_18140-RA	cellulose synthase-like protein G3	Null	1.5
NNU_20039-RA	cellulose synthase A catalytic subunit 2	Null	1.8
NNU_10059-RA	endoglucanase 12-like	Null	−5.2
NNU_07071-RA	xyloglucan glycosyltransferase 4 isoform X1	Null	3.4
**hemicellulose synthesis**			
NNU_01719-RA	putative UDP-glucuronate:xylan alpha-glucuronosyltransferase 3	1.2	Null
^a^ NNU_10542-RA	xyloglucan galactosyltransferase KATAMARI1-like	0.7	−1.7
NNU_12476-RA	probable beta-1,4-xylosyltransferase IRX9 isoform X1	2.0	Null
NNU_13626-RA	probable beta-1,4-xylosyltransferase IRX14H	1.5	Null
**lignin biosynthesis**			
NNU_03759-RA	laccase-4-like	0.7	Null
NNU_03827-RA	phenylalanine ammonia-lyase	1.5	Null
NNU_04966-RA	caffeic acid 3-*O*-methyltransferase 1	1.6	Null
NNU_06036-RA	isoflavone reductase homolog	0.6	Null
NNU_12048-RA	shikimate *O*-hydroxycinnamoyltransferase-like	0.9	Null
NNU_12868-RA	phenylalanine ammonia-lyase-like	1.1	Null
NNU_13598-RA	caffeoyl-CoA *O*-methyltransferase-like	1.0	Null
NNU_14758-RA	4-coumarate--CoA ligase 2-like	1.0	Null
NNU_17055-RA	caffeoyl-CoA *O*-methyltransferase	1.5	Null
NNU_18746-RA	laccase-17-like	0.6	Null
NNU_19318-RA	cinnamoyl-CoA reductase 1-like	1.5	Null
NNU_21321-RA	phenylalanine ammonia-lyase	1.4	Null
NNU_23025-RA	cytochrome P450 98A2	0.8	Null
NNU_23365-RA	cytochrome P450 84A1-like	1.6	Null
^a^ NNU_23877-RA	probable cinnamyl alcohol dehydrogenase 6	0.8	3.6
NNU_24517-RA	cinnamoyl-CoA reductase 2	−0.8	Null
NNU_04106-RA	phenylalanine ammonia-lyase-like	Null	−2.1
NNU_05129-RA	phenylalanine ammonia-lyase-like	Null	−3.6
NNU_07568-RA	cinnamoyl-CoA reductase 2 isoform X2	Null	−1.4
NNU_11085-RA	laccase-17-like	Null	−5.6
NNU_12126-RA	cinnamoyl-CoA reductase 1-like	Null	−5.0
NNU_18647-RA	laccase-7-like	Null	−7.3
NNU_08076-RA	cytochrome P450 84A1-like	Null	1.8
NNU_22838-RA	cinnamoyl-CoA reductase 2-like	Null	2.1
**major CHO metabolism**			
NNU_02421-RA	maltose excess protein 1-like, chloroplastic	0.6	Null
NNU_04943-RA	alkaline/neutral invertase CINV2	1.0	Null
^a^ NNU_17880-RA	probable fructokinase-1	1.2	1.6
^a^ NNU_18248-RA	beta-fructofuranosidase, soluble isoenzyme I-like	0.9	2.3
NNU_19077-RA	sucrose synthase	1.0	Null
NNU_04529-RA	alpha-1,4 glucan phosphorylase L-2 isozyme, chloroplastic/amyloplastic	−0.8	Null
NNU_05767-RA	sucrose synthase 2-like	−1.1	Null
^a^ NNU_11941-RA	beta-fructofuranosidase, insoluble isoenzyme CWINV3-like isoform X1	−0.9	−2.7
NNU_13572-RA	pentatricopeptide repeat-containing protein At2g04860	−1.7	Null
NNU_18912-RA	phosphoglucan phosphatase DSP4, amyloplastic-like	−0.7	Null
NNU_08846-RA	probable fructokinase-7	Null	1.8
NNU_09096-RA	alkaline/neutral invertase CINV2	Null	1.3

Superscript a indicates 11 core genes in DEPs (differentially expressed proteins) and DEGs (differentially expressed genes). Null means proteins or transcripts were not found.

**Table 2 ijms-21-05087-t002:** Proteins involved in six biologic processes were significantly enriched.

Protein Accession	Protein Name	VvsF Ratio	Regulated Type	*p*-Value
**Biosynthesis of amino acids**				
NNU_09156-RA	S-adenosylmethionine synthase 5	2.5	Up	0
NNU_25903-RA	S-adenosylmethionine synthase 2	2.3	Up	0
NNU_07815-RA	5-methyltetrahydropteroyltriglutamate--homocysteine methyltransferase 2-like	2.3	Up	0.001
NNU_06116-RA	glutamine synthetase leaf isozyme, chloroplastic	2.2	Up	0
NNU_15496-RA	pyrroline-5-carboxylate reductase isoform X1	2.2	Up	0.002
NNU_16927-RA	S-adenosylmethionine synthase 1	1.9	Up	0
NNU_22690-RA	phospho-2-dehydro-3-deoxyheptonate aldolase 1, chloroplastic-like	1.9	Up	0
NNU_02525-RA	glutamate synthase 1	1.8	Up	0
NNU_16800-RA	serine acetyltransferase 5-like	1.8	Up	0.003
NNU_18211-RA	3-phosphoshikimate 1-carboxyvinyltransferase 2	1.7	Up	0.001
NNU_21805-RA	5-methyltetrahydropteroyltriglutamate--homocysteine methyltransferase	1.7	Up	0
NNU_18019-RA	serine hydroxymethyltransferase 4	1.7	Up	0
NNU_01496-RA	glutamate synthase 1	1.7	Up	0.031
NNU_13370-RA	phospho-2-dehydro-3-deoxyheptonate aldolase 2, chloroplastic-like	1.7	Up	0
NNU_04572-RA	chorismate mutase 3, chloroplastic-like	1.7	Up	0
NNU_16636-RA	2,3-bisphosphoglycerate-independent phosphoglycerate mutase	1.6	Up	0
NNU_17273-RA	probable fructose-bisphosphate aldolase 3, chloroplastic	1.6	Up	0
NNU_18207-RA	transketolase, chloroplastic	1.6	Up	0
NNU_20134-RA	shikimate kinase, chloroplastic isoform X1	1.6	Up	0
NNU_26622-RA	indole-3-glycerol phosphate synthase, chloroplastic-like isoform X1	1.5	Up	0.015
NNU_01941-RA	aspartokinase 2, chloroplastic-like isoform X1	1.5	Up	0
NNU_14707-RA	D-3-phosphoglycerate dehydrogenase 1, chloroplastic-like	1.5	Up	0
NNU_13158-RA	chorismate synthase, chloroplastic isoform X1	1.5	Up	0.015
NNU_20724-RA	glutamine synthetase cytosolic isozyme 1	1.5	Up	0
NNU_21817-RA	phosphoserine aminotransferase 1, chloroplastic-like	1.5	Up	0
**Flavonoid biosynthesis**				
NNU_16100-RA	naringenin, 2-oxoglutarate 3-dioxygenase-like	4.4	Up	0
NNU_24753-RA	flavonol synthase/flavanone 3-hydroxylase-like	2.7	Up	0.002
NNU_19543-RA	flavonol synthase/flavanone 3-hydroxylase	2.0	Up	0
NNU_12048-RA	shikimate O-hydroxycinnamoyltransferase-like	1.9	Up	0
NNU_04498-RA	flavonoid 3′-monooxygenase-like	1.7	Up	0
NNU_23025-RA	cytochrome P450 98A2	1.7	Up	0
NNU_08856-RA	leucoanthocyanidin dioxygenase-like	1.6	Up	0
**Phenylpropanoid biosynthesis**				
NNU_04966-RA	caffeic acid 3-*O*-methyltransferase 1	3.0	Up	0
NNU_23365-RA	cytochrome P450 84A1-like	3.0	Up	0
NNU_19318-RA	cinnamoyl-CoA reductase 1-like	2.9	Up	0.001
NNU_03827-RA	phenylalanine ammonia–lyase	2.8	Up	0
NNU_21321-RA	phenylalanine ammonia–lyase	2.6	Up	0.001
NNU_12868-RA	phenylalanine ammonia–lyase-like	2.2	Up	0.001
NNU_23881-RA	peroxidase 64	2.0	Up	0
NNU_14758-RA	4-coumarate--CoA ligase 2-like	2.0	Up	0
NNU_12048-RA	shikimate *O*-hydroxycinnamoyltransferase-like	1.9	Up	0
NNU_23025-RA	cytochrome P450 98A2	1.7	Up	0
NNU_23877-RA	probable cinnamyl alcohol dehydrogenase 6	1.7	Up	0.001
NNU_21573-RA	caffeoylshikimate esterase	1.7	Up	0
NNU_03385-RA	peroxidase 42-like	1.6	Up	0.006
NNU_13717-RA	peroxidase 73-like	0.6	Down	0.001
NNU_16422-RA	peroxidase 27-like	0.6	Down	0
NNU_04050-RA	peroxidase P7-like	0.6	Down	0
NNU_20058-RA	peroxidase 12-like	0.6	Down	0
NNU_18799-RA	peroxidase 4-like	0.6	Down	0
NNU_23337-RA	cationic peroxidase 1-like	0.6	Down	0
NNU_04265-RA	peroxidase 27-like	0.6	Down	0.012
NNU_04268-RA	peroxidase 3-like	0.5	Down	0
NNU_22685-RA	peroxidase 17-like	0.5	Down	0
NNU_13190-RA	peroxidase 21-like	0.5	Down	0
NNU_20096-RA	cationic peroxidase 1-like	0.5	Down	0
NNU_02934-RA	peroxidase N-like	0.5	Down	0
NNU_13360-RA	peroxidase 17-like	0.5	Down	0.003
NNU_24553-RA	peroxidase 47-like	0.5	Down	0.001
NNU_12989-RA	peroxidase 2-like	0.5	Down	0
NNU_06410-RA	peroxidase 12-like	0.5	Down	0
NNU_20132-RA	peroxidase 3-like	0.5	Down	0.004
NNU_01736-RA	peroxidase 43-like isoform X1	0.4	Down	0
NNU_20229-RA	peroxidase N1-like	0.4	Down	0
NNU_02064-RA	peroxidase 51	0.4	Down	0
NNU_11196-RA	peroxidase P7-like	0.4	Down	0
NNU_00369-RA	peroxidase 57-like	0.3	Down	0
NNU_13474-RA	peroxidase 10	0.3	Down	0
NNU_21496-RA	aldehyde dehydrogenase family 2 member C4-like	0.3	Down	0
NNU_04048-RA	peroxidase P7-like	0.2	Down	0.003
NNU_12986-RA	peroxidase P7-like	0.2	Down	0
**Amino sugar and nucleotide sugar metabolism**				
NNU_05331-RA	glucose-1-phosphate adenylyltransferase large subunit 1-like	0.7	Down	0.012
NNU_12353-RA	beta-hexosaminidase 1 isoform X1	0.6	Down	0.001
NNU_09609-RA	basic endochitinase-like	0.6	Down	0
NNU_10055-RA	beta-hexosaminidase 3-like	0.6	Down	0
NNU_06174-RA	glucose-1-phosphate adenylyltransferase large subunit 3, chloroplastic/amyloplastic	0.6	Down	0
NNU_10728-RA	acidic endochitinase-like	0.5	Down	0
NNU_12150-RA	G-type lectin S-receptor-like serine/threonine–protein kinase At1g11300	0.5	Down	0
NNU_17908-RA	endochitinase PR4-like	0.5	Down	0
NNU_17907-RA	endochitinase PR4-like	0.4	Down	0
NNU_10867-RA	alpha-l-arabinofuranosidase 1-like	0.4	Down	0
NNU_20770-RA	acidic endochitinase-like	0.4	Down	0
NNU_14306-RA	basic endochitinase-like	0.4	Down	0
NNU_22938-RA	chitinase 5-like, partial	0.3	Down	0
NNU_11632-RA	acidic endochitinase-like	0.3	Down	0
NNU_09610-RA	endochitinase A-like	0.3	Down	0
NNU_17910-RA	endochitinase PR4-like	0.3	Down	0
NNU_12151-RA	acidic mammalian chitinase-like	0.3	Down	0
NNU_22939-RA	chitinase 5-like	0.2	Down	0
NNU_24291-RA	acidic endochitinase-like, partial	0.2	Down	0
**Starch and sucrose metabolism**				
NNU_24463-RA	glucan endo-1,3-beta-glucosidase 6-like	0.7	Down	0.001
NNU_05331-RA	glucose-1-phosphate adenylyltransferase large subunit 1-like	0.7	Down	0.012
NNU_26480-RA	probable sucrose-phosphate synthase 1 isoform X1	0.6	Down	0
NNU_04529-RA	alpha-1,4 glucan phosphorylase L-2 isozyme, chloroplastic/amyloplastic	0.6	Down	0
NNU_06174-RA	glucose-1-phosphate adenylyltransferase large subunit 3, chloroplastic/amyloplastic	0.6	Down	0
NNU_11941-RA	beta-fructofuranosidase, insoluble isoenzyme CWINV3-like isoform X1	0.5	Down	0.001
NNU_07396-RA	beta-glucosidase 40-like	0.5	Down	0
NNU_11617-RA	beta-glucosidase 12-like	0.5	Down	0
NNU_05767-RA	sucrose synthase 2-like	0.5	Down	0
NNU_11613-RA	beta-glucosidase 12-like	0.4	Down	0
NNU_13572-RA	pentatricopeptide repeat-containing protein At2g04860	0.3	Down	0
**Glycosphingolipid biosynthesis**				
NNU_12353-RA	beta-hexosaminidase 1 isoform X1	0.6	Down	0.001
NNU_10055-RA	beta-hexosaminidase 3-like	0.6	Down	0
NNU_08403-RA	alpha-galactosidase-like	0.6	Down	0
NNU_24417-RA	alpha-galactosidase isoform X1	0.5	Down	0

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
