# Peer review of "Integrated Omics Analyses Identify Key Pathways Involved in Petiole Rigidity Formation in Sacred Lotus"

_ijms, 2020, doi:10.3390/ijms21145087_

Round 1
Reviewer 1 Report
Integrating omics analyses in petioles identify key pathways involved in petiole rigidity construction in sacred lotus
Li et al. – ijms 816867
This manuscript reports on a comparative transcriptomics and proteomics study of two kinds of leaves and petioles of the sacred lotus, namely floating and vertical petioles and leaves. A description of the different tissues of the petioles is provided as well as their overall content in cellulose and lignin. Among the identified transcripts and proteins with differential levels of accumulation, some are involved in cell wall biogenesis. The authors conclude that the cell wall plays a role in the differences observed between the petioles of vertical vs floating leaves.
This study is of interest with regard to a puzzling differentiation process. The role of the cell wall could be expected because this is the extracellular skeleton of plants. There are interesting findings in this study, especially the identification of candidate genes regarding the cell wall biogenesis. However, the manuscript is poorly written and the proteomics results need to be checked.
Major comments
The biological material deserves a better description. It is difficult to know at which stage of development it is sampled (lines 447-453). Are the organs (petioles or leaves) still growing or is growth already arrested? Since legend to Fig. S1 is missing, the numbering in red is helpless. Then, it is difficult to assess if the studied physiological process is really the establishment of petiole rigidity as stated in the title. Besides, in Material and Methods, samplinig of leaves is indicated. I did not find any results related to these samples lines (450 and 453).
In the whole manuscript, the authors compare the level of accumulation of transcripts and of proteins. These differential levels of accumulation result from transcriptional to post-translational levels of regulation, including protein degradation. The discrepancy observed for several genes has been already described for genes encoding proteins involved in cell wall biogenesis (see Minic et al. 2009, BMC Plant Biol 9: 17; , Jamet et al. 2009, BMC Genomics 10: 505) (see lines 230-232). Comments like “Transcriptome and proteome analysis revealed that the transcriptional and translational level of several genes involved in … were up-regulated…” should be clarified (lines 600-603).
From Fig. 2, it is difficult to get an overall view of petiole sections. It would be more informative to show complete sections on the left and an enlargement on the right.
Unfortunately, I could not get access to the lotus genomic database and I could not have a look at the identified genes and proteins. Information is missing regarding the way the proteins were identified from MS data. How many specific peptides were considered? It should be at least two. Besides, two tryptic miscliveages have been taken into account. It should be only one. The list of identified proteins might have to be modified.
I have difficulties to understand the content of Tables 1 and 2. In the manuscript, it is mentioned that Among the 135 core genes, 126 have similar trends between the levels of accumulation of their transcripts and proteins (lines 220-221). I was expecting a single table containing information about transcripts and proteins levels. Instead, there are two tables with different contents.
In Table 1 and 2, some genes seem to be wrongly classified:
- How can uncharacterized proteins be classified among cell wall proteins (Table 1)
- Why should xyloglucan endotransglucosylase/Hydrolases (XTH) be classified either as “cell wall degradation” or “cell wall modification”? (Tables 1 and 2)
- - A mannosyltransferase would probably not play roles in cellulose biosynthesis (Tables 1 and 2).
- - What is CHO metabolism?
- - Pectin esterases should be grouped in the “cell wall modification“ group (Tables 1 and 2).
- How can an uncharacterized protein be classified in the “ lignin biosynthesis” group? (Table 2)
As a general comment, the GO terms should be used cautiously. In Table 3, I am not sure that class III peroxidases are involved in the phenylpropanoid metabolism. They are rather involved in the polymerization of monolignols in the cell wall, in the cross-linking of extensins or in the generation of free radicals (see Francoz et al. 2015, Phytochemistry 112: 15). Besides, why should a beta-glucosidase be classified in the “phenylpropanoid biosynthesis” group? Then, why should pectinesterases in the “starch and sucrose metabolism” group?
About the cell wall metabolism, the presence of glycoside hydrolases does not mean that the polysaccharides are degraded (e.g. line 286, line 225, Tables 1 and 2). They could be modified and rearranged (see Frankova and Fry 2013, J Exp Bot 64: 3519). This interpretation should be included.
The experiment in which the PAL inhibitor has been used is not described in Material and Methods. It cannot be provided as a supplementary figure without any explanation (Fig. S7 and lines 325-328). Otherwise, it has to be removed.
Additional comments
The English language should be extensively edited. Even the title includes a mistake: “Integrating… identifies… “. Some incomprehensible sentences or expressions among many others: lines 34-35, lines 81-82, line 91, lines 125-126, line 141, lines 222-223, line 225, lines 306-308, lines 314-316, lines 365-368, line 389, line 442, line 455, … What means “mechanical tissues”, “rigidity construction”?
I guess that the peroxidases mentioned in the manuscript are the class III peroxidases which are secreted in the extracellular space. It has to be checked.
Many figures are fuzzy and difficult to read (Fig. 1, Fig. 2 e-f-g, Fig. 3 g-h, Fig. 4, Fig. 5, Fig. 6, Fig. 7). Figures of better quality should be provided. The arrows on Fig. 2 are hardly visible and the rectangles are missing.
Line 70, ines 417-418: Please, add a reference.
Lines 73-74: Is,’ this study related to a comparison between the petioles of vertical and floating leaves?
Lines 87: What is the difference between cell wall organization and assembly?
Lines 121-122: What does mean “the numbers of air cavities are dependent on petiole development?
Line 147: 2f, 2g instead of 2e, 2f?
Line 219: “misc”?
Line 396: Are flavonoids involved in the formation of lignin?
Line 404-406: I did not see any identified transcription factor in this study. Which metabolites indicate that more lignin is synthesized in vertical petiole?
In Materials and Methods, the city and the country of each purchaser should be indicated everywhere (see line 575). At several places, the names of the purchaser are missing (lines 538, 563, 566, 572). References should be added for the GO annotation method (line 555), the KEGG database (line 555) and the Skyline software (line 567).
The method used to measure the cellulose content is not described (lines 493-497). What is the “cellulose detective apparatus”? (line 493).
Official unit are min for minutes, and s for seconds. Please modify the text accordingly.
Each abbreviation should be clearly defined at its first appearance in the text and the list provided p. 22 should be in alphabetical order.
The reference list should be extensively edited, so that all the references are presented in the same way. Many references are incomplete: e.g. ref. 1, 3, 11, 12, 13, 17, 21, 22, 29.
Typos: glycosyltransferases (line 52), Physiology (line 635).
The supplementary files have no title and no legend.
Author Response
Revierer 1
This manuscript reports on a comparative transcriptomics and proteomics study of two kinds of leaves and petioles of the sacred lotus, namely floating and vertical petioles and leaves. A description of the different tissues of the petioles is provided as well as their overall content in cellulose and lignin. Among the identified transcripts and proteins with differential levels of accumulation, some are involved in cell wall biogenesis. The authors conclude that the cell wall plays a role in the differences observed between the petioles of vertical vs floating leaves.
This study is of interest with regard to a puzzling differentiation process. The role of the cell wall could be expected because this is the extracellular skeleton of plants. There are interesting findings in this study, especially the identification of candidate genes regarding the cell wall biogenesis. However, the manuscript is poorly written and the proteomics results need to be checked.
Major comments
The biological material deserves a better description. It is difficult to know at which stage of development it is sampled (lines 447-453). Are the organs (petioles or leaves) still growing or is growth already arrested? Since legend to Fig. S1 is missing, the numbering in red is helpless. Then, it is difficult to assess if the studied physiological process is really the establishment of petiole rigidity as stated in the title. Besides, in Material and Methods, samplinig of leaves is indicated. I did not find any results related to these samples lines (450 and 453).
Response: Thank you very much for your comments. Speaking of the stage of petioles development, it has a significant change during growth. No matter it is floating leaf or vertical leaf, the leaf is folded at the beginning of growth, and then the leaf will unfold at last. We sample the petioles at the stage with folded leaf, and it means the petioles is still growing. Even though the petiole is growing, the two types of petioles are formed. Considering the expression levels of transcripts and proteins might come back to normal when the leaf no longer grows, we sample petioles at the early stage of two types petioles formed. As for sampling of leaves, it is a mistake. We have checked some genes transcripts levels using PCR, but the results were not used at last. We have revised related parts in results and added the supplementary files to make text far from misunderstanding.
In the whole manuscript, the authors compare the level of accumulation of transcripts and of proteins. These differential levels of accumulation result from transcriptional to post-translational levels of regulation, including protein degradation. The discrepancy observed for several genes has been already described for genes encoding proteins involved in cell wall biogenesis (see Minic et al. 2009, BMC Plant Biol 9: 17; , Jamet et al. 2009, BMC Genomics 10: 505) (see lines 230-232). Comments like “Transcriptome and proteome analysis revealed that the transcriptional and translational level of several genes involved in … were up-regulated…” should be clarified (lines 600-603).
Response: Thank you very much for your suggestions. In our work, we compare the changes of abundance of transcripts and proteins in two types of petioles respectively first, and then we try to compare the results from two omics data. We have made discussions relating to the previous work in your suggested literature in Discussion part.
From Fig. 2, it is difficult to get an overall view of petiole sections. It would be more informative to show complete sections on the left and an enlargement on the right.
Response: Yes, thanks for your suggestions. We have revised Fig2 as your suggestion.
Unfortunately, I could not get access to the lotus genomic database and I could not have a look at the identified genes and proteins. Information is missing regarding the way the proteins were identified from MS data. How many specific peptides were considered? It should be at least two. Besides, two tryptic miscliveages have been taken into account. It should be only one. The list of identified proteins might have to be modified.
Response: Thanks for your comments. We are sorry that the original lotus database(http://lotus-db.wbgcas.cn/ Wang et al.,2015 Database) was crashed. The data was collected in some other database now such as PLAZA 4.0 Dicot (https://bioinformatics.psb.ugent.be/plaza/versions/plaza_v4_dicots/). We have revised it in manuscript.
Yes. You are right. Specific peptides set to two and tryptic miscliveage sets to one in protein identification were stricter standard. However, our work was started and finished in the end of 2016. At that time, we consult the general standards on others articles and used in our work (RogerGeiger, et al., Cell, 2016).
I have difficulties to understand the content of Tables 1 and 2. In the manuscript, it is mentioned that Among the 135 core genes, 126 have similar trends between the levels of accumulation of their transcripts and proteins (lines 220-221). I was expecting a single table containing information about transcripts and proteins levels. Instead, there are two tables with different contents.
Response: We are sorry to mislead you on understanding the relation of core gene and genes in table 1 and table 2. First, the levels of accumulation of the 135 core genes were showed in revised Fig. S4b. Among the 135 core genes, 126 have similar trends between the levels of accumulation of their transcripts and proteins (see Fig. S4b). Genes and proteins in Table 1 and Table 2 are DEGs (differentially expressed genes between IVP and IFP) and DEPs (differentially expressed proteins between IVP and IFP) involved in cell wall biosynthesis. Only 11 core genes were identified in Table 1 and Table 2. Then we have combined the contents of table 1 and table 2 into a new table.
In Table 1 and 2, some genes seem to be wrongly classified:
- How can uncharacterized proteins be classified among cell wall proteins (Table 1)
Response: Thanks for your comments. First, the name of these genes or proteins come from lotus genome sequencing and annotation. In table1 and table 2, function categories of proteins or genes were predicted using MAPMAN software(Oliver Thimm, et al., The plant journal, 2004). The basic principle is that candidate proteins or genes were blast to all genes of plant which have clear annotation. In this case, it might because these four uncharacterized proteins have similar domain with other proteins which have functions in cell wall related. However, in order to make results more accurate, these four uncharacterized proteins were deleted from tables.
- Why should xyloglucan endotransglucosylase/Hydrolases (XTH) be classified either as “cell wall degradation” or “cell wall modification”? (Tables 1 and 2)
Response: Yes. It is our mistakes. We have revised it in new table and text.
- - A mannosyltransferase would probably not play roles in cellulose biosynthesis (Tables 1 and 2).
Response: Yes, you are right. In fungi, mannosyltransferase was reported to contribute to cell wall integrity or cell wall biosynthesis (Tingting Zhao,et al., Current genetics, 2019; Shaun M. Bowman, et al., Mycologia, 2005). As revision, these two mannosyltransferase were classified into cell wall protein group.
- - What is CHO metabolism?
Response: It is the second BIN desigination in MAPMAN software, genes in this group are involved in carbohydrates metabolism such as sucrose and starch metabolism (Oliver Thimm, et al., The plant journal, 2004).
- - Pectin esterases should be grouped in the “cell wall modification“ group (Tables 1 and 2).
Response: Thank you for your suggestion. It has been revised.
- How can an uncharacterized protein be classified in the “ lignin biosynthesis” group? (Table 2)
Response: Thanks for your comments. First, the name of these genes or proteins come from lotus genome sequencing and annotation. In table1 and table 2, function categories of proteins or genes were predicted using MAPMAN software(Oliver Thimm, et al., The plant journal, 2004). The basic principle is that candidate proteins or genes were blast to all genes of plant which have clear annotation. In this case, it might because these four uncharacterized proteins have similar domain with other proteins which have functions in cell wall related. However, in order to make results more accurate, these four uncharacterized proteins were deleted from tables.
As a general comment, the GO terms should be used cautiously. In Table 3, I am not sure that class III peroxidases are involved in the phenylpropanoid metabolism. They are rather involved in the polymerization of monolignols in the cell wall, in the cross-linking of extensins or in the generation of free radicals (see Francoz et al. 2015, Phytochemistry 112: 15). Besides, why should a beta-glucosidase be classified in the “phenylpropanoid biosynthesis” group? Then, why should pectinesterases in the “starch and sucrose metabolism” group?
Response: Yes. You are right. There are some mistakes in the using of GO terms. In KEGG pathway, class III peroxidases are involved in the phenylpropanoid biosynthesis. In revised table 2(Table 3 in last vision manuscript), these peroxidases are enriched in phenylpropanoid biosynthesis not phenylpropanoid metabolism. Beta-glucosidase was removed from the “phenylpropanoid biosynthesis” group and pectinesterases were removed from the “starch and sucrose metabolism” group.
About the cell wall metabolism, the presence of glycoside hydrolases does not mean that the polysaccharides are degraded (e.g. line 286, line 225, Tables 1 and 2). They could be modified and rearranged (see Frankova and Fry 2013, J Exp Bot 64: 3519). This interpretation should be included.
Response: Thanks for your suggestion. We have added some interpretation in relevant parts.
The experiment in which the PAL inhibitor has been used is not described in Material and Methods. It cannot be provided as a supplementary figure without any explanation (Fig. S7 and lines 325-328). Otherwise, it has to be removed.
Response: Yes. It was removed.
Additional comments
The English language should be extensively edited. Even the title includes a mistake: “Integrating… identifies… “. Some incomprehensible sentences or expressions among many others: lines 34-35, lines 81-82, line 91, lines 125-126, line 141, lines 222-223, line 225, lines 306-308, lines 314-316, lines 365-368, line 389, line 442, line 455, … What means “mechanical tissues”, “rigidity construction”?
Response: Thanks for your comments. Manuscript has been edited by native English speaker now.
I guess that the peroxidases mentioned in the manuscript are the class III peroxidases which are secreted in the extracellular space. It has to be checked.
Response: Yes, you are right. Class III peroxidases are involved in cell wall biosynthesis.
Many figures are fuzzy and difficult to read (Fig. 1, Fig. 2 e-f-g, Fig. 3 g-h, Fig. 4, Fig. 5, Fig. 6, Fig. 7). Figures of better quality should be provided. The arrows on Fig. 2 are hardly visible and the rectangles are missing.
Response: Thanks for your comments. We have replaced the figure with high-resolution ones.
Line 70, ines 417-418: Please, add a reference.
Response: Thank you. We have added them.
Lines 73-74: Is,’ this study related to a comparison between the petioles of vertical and floating leaves?
Response: Yes, you are right. We compared proteins and transcripts expressed in two types of petioles.
Lines 87: What is the difference between cell wall organization and assembly?
Response: Usually, cell wall organization and cell wall assembly were used without distinction. However, there are some fine distinction. If we talk about cell wall assembly, to a considerable extent, we are discussing about the assembling of CESA monomers into rosette complexes, or the assembling the cellulose/xyloglucan network. The cell wall organization is more general than cell wall assembly. It means how cell use cellulose, hemicelluloses, pectic, lignin and cell wall structural proteins to build the full function cell wall (Book: Plant Cell Separation and Adhesion, Chapter 2, 2007; Book: Molecular Cell Biology of the Growth and Differentiation of Plant Cells, Chapter 7, 2016).
Lines 121-122: What does mean “the numbers of air cavities are dependent on petiole development?
Response: Sorry. We have rewritten this sentence.
Line 147: 2f, 2g instead of 2e, 2f?
Response: Thanks for your correction. We have done it.
Line 219: “misc”?
Response: It is a It is a BIN desigination in MAPMAN software(Oliver Thimm, et al., The plant journal, 2004), genes in this group are involved in varies biochemical reaction. Many enzymes such as lipase, transferases, lyases and others will be classified in this group.
Line 396: Are flavonoids involved in the formation of lignin?
Response: May be not. But cinnamoyl-CoA is in the upstream of phenylpropanoid biosynthesis pathway. It can be used in the formation of lignin, and also can be used in flavonoids biosynthesis (Benoît van der Rest, et al., Journal of Experimental Botany 2006; Se´bastien Besseau, et al., The Plant Cell, 2007).
Line 404-406: I did not see any identified transcription factor in this study. Which metabolites indicate that more lignin is synthesized in vertical petiole?
Response: TFs identified in proteomics and transcriptomics were listed in supplementary table 5. As for which metabolites indicate that more lignin is synthesized in vertical petiole. Frankly speaking, we have not checked the abundance of syringyl lignin monomers, hydroxyphenyl lignin monomers, and guaiacyl lignin monomers, but the precursors abundance of these three lignin monomers were checked. The precursor abundance of syringyl lignin monomers and hydroxyphenyl lignin monomer were significantly higher in IVP compared to IFP. The precursor abundance of guaiacyl lignin monomers are same in IVP and IFP. Considering the higher lignin content in IVP than IFP (Fig. 2), we speculate there might more lignin is synthesized in vertical petiole.
In Materials and Methods, the city and the country of each purchaser should be indicated everywhere (see line 575). At several places, the names of the purchaser are missing (lines 538, 563, 566, 572). References should be added for the GO annotation method (line 555), the KEGG database (line 555) and the Skyline software (line 567).
Response: Thanks for your correction. We have revised all of them.
The method used to measure the cellulose content is not described (lines 493-497). What is the “cellulose detective apparatus”? (line 493).
Response: we have added the methods of cellulose and lignin content measurement.
Official unit are min for minutes, and s for seconds. Please modify the text accordingly.
Response: Thanks for your correction. We have revised all of them in manscript.
Each abbreviation should be clearly defined at its first appearance in the text and the list provided p. 22 should be in alphabetical order.
Response: Yes. We have revised it.
The reference list should be extensively edited, so that all the references are presented in the same way. Many references are incomplete: e.g. ref. 1, 3, 11, 12, 13, 17, 21, 22, 29.
Response: Thanks for your careful check. We have checked and revised all references.
Typos: glycosyltransferases (line 52), Physiology (line 635).
Response: Thanks for your correction. It has been revised.
The supplementary files have no title and no legend.
Response: Thanks for your correction. It has been revised.
Reviewer 2 Report
Comments to the Authors
The authors report about study of petiole rigidity construction using omics approaches. Authors assessed the difference from anatomical structure to the expression of genes and proteins in two petioles types of lotus differing in rigidity. They stated that the results are going to help in better understanding of the mechanism of cell wall formation. That I consider as interesting general purpose of the study. I suggest to elaborate a little more on this idea in the Discussion. Summarising, the idea of these broad study is interesting, results are clearly presented, discussion is conducted generally in good way, the conclusion are adequate to the achieved data. The manuscript is well described and I consider it as being almost ready for the publication.
Remark
L. 383. Remove ‘, which’ from the sentence, that will make it correct.
Author Response
Reviewer 2
Comments to the Authors
The authors report about study of petiole rigidity construction using omics approaches. Authors assessed the difference from anatomical structure to the expression of genes and proteins in two petioles types of lotus differing in rigidity. They stated that the results are going to help in better understanding of the mechanism of cell wall formation. That I consider as interesting general purpose of the study. I suggest to elaborate a little more on this idea in the Discussion. Summarising, the idea of these broad study is interesting, results are clearly presented, discussion is conducted generally in good way, the conclusion are adequate to the achieved data. The manuscript is well described and I consider it as being almost ready for the publication.
Remark
- 383. Remove ‘, which’ from the sentence, that will make it correct.
Response: Thank you very much for your comments. It has been revised.